# Covariance-Matrix-Based Criteria for Network Entanglement

**DOI:** 10.3390/e25091260

**Published:** 2023-08-24

**Authors:** Kiara Hansenne, Otfried Gühne

**Affiliations:** Naturwissenschaftlich-Technische Fakultät, Universität Siegen, Walter-Flex-Straße 3, 57068 Siegen, Germany

**Keywords:** quantum networks, network entanglement, covariance matrices

## Abstract

Quantum networks offer a realistic and practical scheme for generating multiparticle entanglement and implementing multiparticle quantum communication protocols. However, the correlations that can be generated in networks with quantum sources and local operations are not yet well understood. Covariance matrices, which are powerful tools in entanglement theory, have been also applied to the network scenario. We present simple proofs for the decomposition of such matrices into the sum of positive semi-definite block matrices and, based on that, develop analytical and computable necessary criteria for preparing states in quantum networks. These criteria can be applied to networks where nodes share at most one source, such as all bipartite networks.

## 1. Introduction

Entanglement is a central element in quantum theory and the subject of famous debates at the beginning of the 20th century [1,2] and gained the status of a resource with the advent of quantum information theory some decades later (see Refs. [3,4] for reviews). Entanglement between two parties has been widely studied and characterised, but much less is known regarding multiparticle entanglement. Indeed, when more than two parties are involved, the structure of entanglement becomes more complex, with non-equivalent classes of entanglement appearing [3,4]. Apart from the foundational interest in understanding the structure of multiparticle entanglement, the significance lies in the fact that it also is a resource for many quantum information applications, such as quantum conference key distribution [5], quantum error correcting codes [6], or high-precision metrology [7]. However, generating and manipulating genuine multipartite entangled states experimentally is a difficult task, particularly when the number of entangled parties is large (see Ref. [4] and references therein). To circumvent this issue, the arguably more experimentally friendly concept of quantum networks has been introduced [8,9]. In the network setup, the goal is to generate a global *N*-partite state using a set of sources (represented by edges in a (hyper)graph) that distribute (connect) subsystems of entangled states to the different parties of the network (the nodes of the (hyper)graph). Strictly, we require sources to distribute particles to at most N−1 nodes, and the parties might be allowed to apply a local operation to their system. Figure 1 details an example of a tripartite network with bipartite sources.

The power and limitations of such networks have already been studied in Refs. [10,11,12,13,14,15,16,17,18,19], however, it is still unclear which useful quantum states can actually be prepared using them. In the most general definition, the parties and the sources can additionally share a global classical random variable, and we say that such networks arise from local operations and shared randomness (losr). However, it is also realistic to consider models where there is no access to such a global variable. The main aim of this paper is to show that covariance matrices (cms) can be used to derive strong criteria for entanglement in the various network scenarios.

First introduced for continuous variable systems [20,21], covariance matrices possess useful properties and have previously been used to characterise bipartite and multiparticle entanglement [22,23]. Recently, they have also been used to derive necessary criteria for network scenarios [11,14,24,25]. In Ref. [11], the authors formulate a necessary condition for a probability distribution to arise from measurements performed on a quantum network state. The condition states that the covariance matrix of the probability distribution can be decomposed into the sum of positive semi-definite (psd) block matrices, and can be formulated as a semi-definite program (sdp). We call this decomposition into a sum of psd block matrices the block decomposition of a covariance matrix. This result was applied in Ref. [14] to derive practical analytical criteria for networks with dichotomic measurements and for networks with bipartite sources. More recently, similar sdps were developed in Ref. [25] for the case of losr networks, with extra assumptions on rank and purity. Finally, aiming at generality, the authors of Ref. [24] showed that in the case of no-common-double-source (ncds) networks (in no-common-double-source networks, two nodes can hold subsystems from at most one common source), the block decomposition criterion holds for all generalised probabilistic theories.

In this paper, we propose an alternative proof to the block decomposition of the cm of the triangle network state derived in Ref. [11]. From it, we obtain an analytical, computable necessary criterion for a state to arise from a triangle network. This criterion can also be used to upper-bound the maximal fidelity a triangle network state can have to a given target state, for instance the GHZ state. We discuss the fact that these bounds are still valid for networks with losr and, finally, show how this result can be extended to ncds networks.

## 2. Network Entanglement

The general triangle network situation involves Alice, Bob, and Charlie wanting to share a tripartite (entangled) state, but they only have access to bipartite sources, as shown in Figure 1. This situation differs from the usual consideration of tripartite entanglement, where the parties have access to a tripartite state generated by a global source. In addition, the parties in the triangle network considered here do not have access to classical communication, which prevents them from executing teleportation or entanglement swapping protocols. Although classical communication is usually considered a cheap resource in quantum communication protocols, it does require time. The classical information must be communicated across the network, which can introduce undesirable latency, particularly in a context where quantum memories are still sub-optimal and expensive.

In this manuscript, we will focus on different triangle network scenarios: the basic triangle network (btn), where bipartite sources are shared among the parties; the triangle network with local unitaries (utn), where Alice, Bob, and Charlie are allowed to perform unitary operations on their local systems; and finally, the triangle network with local channels (ctn), where, as the name indicates, local channels are performed by the parties.

In the btn, three (entangled) bipartite source states ($\varrho_{a}$, $\varrho_{b}$, and $\varrho_{c}$) are prepared and each subsystem is sent to a node according to the distribution in Figure 1. Alice, Bob, and Charlie own the bipartite systems $A_{1}A_{2}=A$, $B_{1}B_{2}=B$, and $C_{1}C_{2}=C$, respectively. The global state of the system ABC reads
(1)$$\varrho_{\mathrm{BTN}}=\varrho_{b}⊗\varrho_{c}⊗\varrho_{a}.$$
Notice that the order of the subsystems is not ABC for the right-hand side, it is organised following the partition $C_{2}A_{1}A_{2}B_{1}B_{2}C_{1}$. The reduced states of Alice, Bob, and Charlie are separable bipartite states. This scenario has, for instance, been studied in the context of pair entangled network states [16]. In this work, we will assume that the sources all send d×d-dimensional states, while keeping in mind that all the results can easily be extended to unequal dimensions.

In the following two scenarios, we allow the parties to perform operations on their local systems. First, we only give Alice, Bob, and Charlie the possibility of performing a unitary operation on their system, namely, $U_{A}$, $U_{B}$, and $U_{C}$, respectively. This leads to the following global state
(2)$$\varrho_{\mathrm{UTN}}=\left(U_{A}⊗U_{B}⊗U_{C}\right)\varrho_{\mathrm{BTN}}\left(U_{A}^{†}⊗U_{B}^{†}⊗U_{C}^{†}\right).$$
Alice, Bob, and Charlie no longer necessarily hold separable bipartite states. We note that here again, there is no tripartite interaction between the parties. Second, we drop the unitary restriction on the local operations, meaning that Alice, Bob, and Charlie may now apply channels on their local systems, represented by completely positive and trace preserving maps $E_{A}$, $E_{B}$, and $E_{C}$, respectively. In that case, the global state reads
(3)$$\varrho_{\mathrm{CTN}}=E_{A}⊗E_{B}⊗E_{C}\left(\varrho_{\mathrm{BTN}}\right).$$
We note that if the dimensions match, then $\left\{\varrho_{\mathrm{BTN}}\right\}\subset \left\{\varrho_{\mathrm{UTN}}\right\}\subset \left\{\varrho_{\mathrm{CTN}}\right\}$, but in general ctn networks can be defined in broader scenarios, since the maps $E_{A}$, $E_{B}$, and $E_{C}$ may reduce the dimension.

These definitions naturally extend to networks with a higher number of parties or sources. A special instance is the previously-mentioned ncds networks: in this case, any two parties share subsystems from at most one source. For instance, all bipartite networks are ncds.

Finally, we could also allow the whole system to be coordinated by a global classical random variable λ. In the most general situation, this would result in states of the form $\varrho_{\Delta }=\sum_{\lambda }p_{\lambda }\varrho_{\mathrm{CTN}}^{\left(\lambda \right)}$. These networks are called losr networks. One direct consequence is that the set of states $\left\{\varrho_{\Delta }\right\}$ is convex, whereas Equations (Equation 1)–(Equation 3) lead to non-convex state sets. As already pointed out in Refs. [10,17], in the case of unbounded source dimensions, it suffices to consider that either the state or the parties have sole access to the global variable.

## 3. Covariance Matrices

In this paper, the tools used to analyse network entanglement are covariance matrices, which characterise states through the covariance of some given observables. In practice, the cm Γ is constructed for a state ϱ and a set of observables $\left\{M_{i}\right\}$, and has the following matrix elements
(4)$${\left[\Gamma \left(\left\{M_{i}\right\},\varrho \right)\right]}_{mn}={\langle M_{m}M_{n}\rangle }_{\varrho }-{\langle M_{m}\rangle }_{\varrho }{\langle M_{n}\rangle }_{\varrho }$$
with ${\langle X\rangle }_{\varrho }=\left(X\varrho \right)$ being the expectation value of the observable *X* when the state of the system is given by ϱ. As in network scenarios the parties can only access their local systems, it is sensible to choose observables $A_{i}$, $B_{j}$, and $C_{k}$ that only act on Alice’s, Bob’s, and Charlie’s sides, respectively. Explicitly, we have $A_{i}⊗1_{B}⊗1_{C}$, $1_{A}⊗B_{j}⊗1_{C}$, and $1_{A}⊗1_{B}⊗C_{k}$, and we will use the notation $\left\{A_{i},B_{j},C_{k}\right\}={\left\{A_{i}⊗1_{B}⊗1_{C}\right\}}_{i}\cup {\left\{1_{A}⊗B_{j}⊗1_{C}\right\}}_{j}\cup {\left\{1_{A}⊗1_{B}⊗C_{k}\right\}}_{k}$. In that case, the cm of a tripartite state ϱ has the following block structure:(5)$$\Gamma \left(\left\{A_{i},B_{j},C_{k}\right\},\varrho \right)=\left(\begin{array}{ccc} \Gamma_{A} & \gamma_{E} & \gamma_{F}\\ \gamma_{E}^{T} & \Gamma_{B} & \gamma_{G}\\ \gamma_{F}^{T} & \gamma_{G}^{T} & \Gamma_{C} \end{array}\right)$$
where $\Gamma_{A}=\Gamma \left(\left\{A_{i}\right\},\varrho^{\left(A\right)}\right)$ is the cm of the reduced state $\varrho^{\left(A\right)}$. For a state ϱ on a system XY, we denote by ${\mathrm{tr}}_{Y}\left(\varrho \right)=\varrho^{\left(X\right)}$ the reduced state of the subsystem *X*. The matrices $\Gamma_{B}$ and $\Gamma_{C}$ have analogous expressions. The elements of the off-diagonal block $\gamma_{E}$ are given by the real numbers
(6)$${\left[\gamma_{E}\right]}_{mn}={\langle A_{m}⊗B_{n}\rangle }_{\varrho }-{\langle A_{m}\rangle }_{\varrho }{\langle B_{n}\rangle }_{\varrho },$$
with identity operators padded where needed (note that Equation (Equation 6) can be defined equivalently by taking the expectation values on $\varrho^{\left(AB\right)}$). Again, the matrices $\gamma_{F}$ and $\gamma_{G}$ can be expressed in a similar way.

## 4. Basic Triangle Network

In this section, we derive the explicit structure of cms of btn states. Let us first define what we will call the reduced observable $A_{i}^{\left(2\right)}$ of $A_{i}$, which describes an effective observable on the system $A_{2}.$ It is given by
(7)$$\begin{array}{cc} A_{i}^{\left(2\right)} & ={\mathrm{tr}}_{A_{1}}\left(A_{i}\left[\varrho_{\mathrm{BTN}}^{\left(A_{1}\right)}⊗1_{A_{2}}\right]\right). \end{array}$$
Note that $A_{i}$ acts on both $A_{1}$ and $A_{2}$, so $A_{i}^{\left(2\right)}$ is an operator acting on states of $A_{2}$, where the effect of $\varrho_{b}=\varrho_{\mathrm{BTN}}^{\left(A_{1}C_{2}\right)}$ has been taken into account. We define $B_{j}^{\left(1\right)}$ similarly and will use the notation $\left\{A_{i}^{\left(2\right)},B_{j}^{\left(1\right)}\right\}={\left\{A_{i}^{\left(2\right)}⊗1_{B_{1}}\right\}}_{i}\cup {\left\{1_{A_{2}}⊗B_{j}^{\left(1\right)}\right\}}_{j}$. The off-diagonal blocks of Equation (Equation 5) can be expressed using the reduced observables, that is,
(8)$${\left[\gamma_{E}\right]}_{mn}=\langle A_{m}^{\left(2\right)}⊗B_{n}^{\left(1\right)}\rangle -\langle A_{m}^{\left(2\right)}\rangle \langle B_{n}^{\left(1\right)}\rangle .$$
To see this, we notice that the reduces state $\varrho_{\mathrm{BTN}}^{\left(AB\right)}$ is a product state with respect to the partition $A_{1}|A_{2}B_{1}|B_{2}$ and use a local basis decomposition of the observables $A_{m}$ and $B_{n}$ (see Appendix A). All expectation values of Equation (Equation 8) are taken with respect to the state $\varrho_{\mathrm{BTN}}^{\left(A_{2}B_{1}\right)}$, which is nothing but $\varrho_{c}$.

This representation means that $\gamma_{E}$ can be computed using only the reduced observables on the state $\varrho_{\mathrm{BTN}}^{\left(A_{2}B_{1}\right)}$. This is a direct consequence of the fact that the marginal states of Alice, Bob, and Charlie are product states, which will no longer be the case in the next scenarios. Let us now introduce our first proposition.

**Proposition** **1**(Block decomposition for cms of btn states)**.**
*The* cm *of a* btn *state with local observables $\left\{A_{i},B_{j},C_{k}\right\}$ can be decomposed as*
(9)$$\begin{array}{cc} \Gamma_{\mathrm{BTN}}= & \Gamma (\left\{A_{i},B_{j},C_{k}\right\},\varrho_{\mathrm{BTN}})\\ = & \underset{T_{c}}{\underset{︸}{\left(\begin{array}{ccc} \Gamma_{A_{2}} & \gamma_{E} & 0\\ \gamma_{E}^{T} & \Gamma_{B_{1}} & 0\\ 0 & 0 & 0 \end{array}\right)}}+\underset{T_{b}}{\underset{︸}{\left(\begin{array}{ccc} \Gamma_{A_{1}} & 0 & \gamma_{F}\\ 0 & 0 & 0\\ \gamma_{F}^{T} & 0 & \Gamma_{C_{2}} \end{array}\right)}}+\underset{T_{a}}{\underset{︸}{\left(\begin{array}{ccc} 0 & 0 & 0\\ 0 & \Gamma_{B_{2}} & \gamma_{G}\\ 0 & \gamma_{G}^{T} & \Gamma_{C_{1}} \end{array}\right)}}+\underset{R}{\underset{︸}{\left(\begin{array}{ccc} R_{A} & 0 & 0\\ 0 & R_{B} & 0\\ 0 & 0 & R_{C} \end{array}\right)}} \end{array}$$
*where the matrices*
$T_{a}$, $T_{b}$*, and*
$T_{c}$
*are* cm*s for the state-dependent reduced observables, i.e.,*
(10)$$T_{c}=\Gamma \left(\left\{A_{i}^{\left(2\right)},B_{j}^{\left(1\right)}\right\},\varrho_{\mathrm{BTN}}^{\left(A_{2}B_{1}\right)}\right).$$
*and analogously for $T_{b}$ and $T_{a}$. The matrix R is positive semi-definite.*

Using Equation (Equation 8), it is only left to show that $R_{A}=\Gamma_{A}-\Gamma_{A_{1}}-\Gamma_{A_{2}}$ is psd, as well as $R_{B}$ and $R_{C}$. To achieve this, we show that 〈$x|R_{A}|x$〉 can always be written as the trace of a product of psd matrices. The proof is given in Appendix B.

We want to emphasize that the results presented in this manuscript are valid only in the context of finite-dimensional Hilbert spaces. A potential future research direction is to investigate how these results can be extended to the infinite-dimensional case. As mentioned in the introduction, cms are also well suited for continuous variable systems.

Armed with this, we can now derive the structure of the covariance matrix of a btn state when the observables are full sets of local orthogonal observables, namely, $\left\{A_{i}\right\}=\left\{G_{\alpha }^{\left(A_{1}\right)}⊗G_{\beta }^{\left(A_{2}\right)}\right\}$, where $\left\{G_{\alpha }^{\left(A_{k}\right)}\right\}$ is a set of $d^{2}$ orthogonal observables acting on states of $A_{k}$ such that $\left(G_{\alpha }^{\left(A_{k}\right)}G_{\alpha^{\prime }}^{\left(A_{k}\right)}\right)=d\delta_{\alpha \alpha^{\prime }}$ (k=1,2). This is performed in a similar way for the systems *B* and *C*. When the situation is explicit enough, we will drop the superscripts. In the case of qubits, the Pauli operators $\sigma_{x}$, $\sigma_{y}$, and $\sigma_{z}$, together with the 2×2 identity operator 1, are an obvious choice. With such sets of observables, a direct computation (see Appendix C) shows that
(11)$$\begin{array}{cc} R_{X}= & \Gamma_{X}-\Gamma_{X_{1}}-\Gamma_{X_{2}}\\ = & \Gamma \left(\left\{G_{\alpha }\right\},\varrho_{\mathrm{BTN}}^{\left(X_{1}\right)}\right)⊗\Gamma \left(\left\{G_{\beta }\right\},\varrho_{\mathrm{BTN}}^{\left(X_{2}\right)}\right),\;X=A,B,C \end{array}$$
and, therefore, *R* is trivially psd in the case of full sets of orthogonal observables.

The structure of the matrices $T_{a}$, $T_{b}$, and $T_{c}$ can also be further explored. First, let us compute the reduced observables
(12)$$A_{i}^{\left(2\right)}=\mathrm{tr}(G_{\alpha }\varrho_{\mathrm{BTN}}^{\left(A_{1}\right)})G_{\beta }=a_{\alpha }^{\left(1\right)}G_{\beta }.$$
where the coefficients $a_{\alpha }^{\left(1\right)}=\mathrm{tr}(G_{\alpha }\varrho_{\mathrm{BTN}}^{\left(A_{1}\right)})$ are nothing but the (real) Bloch coefficients of the reduced states. In Appendix D, we show that
(13)$$\Gamma_{A_{2}}=|{\vec{a}}^{\left(1\right)}\rangle \langle {\vec{a}}^{\left(1\right)}|⊗\Gamma \left(\left\{G_{\beta }\right\},\varrho_{\mathrm{BTN}}^{\left(A_{2}\right)}\right)$$
and that
(14)$$\gamma_{E}=|{\vec{a}}^{\left(1\right)}\rangle \langle {\vec{b}}^{\left(2\right)}|⊗\gamma \left(\left\{G_{\beta },G_{\alpha }\right\},\varrho_{\mathrm{BTN}}^{\left(A_{2}B_{1}\right)}\right)$$
with $|{\vec{a}}^{\left(1\right)}\rangle ={\left(a_{0}^{\left(1\right)},\ldots ,a_{d^{2}-1}^{\left(1\right)}\right)}^{T}\in R^{d^{2}}$ and similarly for $|{\vec{b}}^{\left(2\right)}\rangle$. The matrix $\gamma (\left\{G_{\beta },G_{\alpha }\right\},\varrho_{\mathrm{BTN}}^{\left(A_{2}B_{1}\right)})$ is the off-diagonal block of the cm with the same observables and state. Finally, we can write
(15)$$T_{c}=|{\vec{a}}^{\left(1\right)}⊕{\vec{b}}^{\left(2\right)}\rangle \langle {\vec{a}}^{\left(1\right)}⊕{\vec{b}}^{\left(2\right)}|★\Gamma \left(\varrho_{\mathrm{BTN}}^{\left(A_{2}B_{1}\right)}\right),$$
where ★ is the “block-wise” Kronecker product, called the Khatri–Rao product [26,27]. Formally, if *A* and *B* are block matrices, the i,jth block of their Khatri–Rao product, ${\left(A★B\right)}_{i,j}$, is the Kronecker product of the i,jth block of *A* and *B*, $A_{i,j}⊗B_{i,j}$. For instance, if *A* and *B* are 2×2 block matrices,
(16)$$A=\left(\begin{array}{cc} A_{0,0} & A_{0,1}\\ A_{1,0} & A_{1,1} \end{array}\right),\;B=\left(\begin{array}{cc} B_{0,0} & B_{0,1}\\ B_{1,0} & B_{1,1} \end{array}\right),$$
we obtain
(17)$$A★B=\left(\begin{array}{cc} A_{0,0}⊗B_{0,0} & A_{0,1}⊗B_{0,1}\\ A_{1,0}⊗B_{1,0} & A_{1,1}⊗B_{1,1} \end{array}\right)$$
(see Ref. [27] for more details).

Finally, one has

**Proposition** **2.***The* cm *of a* btn *state, using complete sets of orthogonal observables acting locally can be decomposed as*(18)$$\begin{array}{cc} \Gamma_{\mathrm{BTN}}= & |{\vec{a}}^{\left(1\right)}⊕{\vec{b}}^{\left(2\right)}\rangle \langle {\vec{a}}^{\left(1\right)}⊕{\vec{b}}^{\left(2\right)}|★\Gamma \left(\varrho_{\mathrm{BTN}}^{\left(A_{2}B_{1}\right)}\right)+|{\vec{b}}^{\left(1\right)}⊕{\vec{c}}^{\left(2\right)}\rangle \langle {\vec{b}}^{\left(1\right)}⊕{\vec{c}}^{\left(2\right)}|★\Gamma \left(\varrho_{\mathrm{BTN}}^{\left(B_{2}C_{1}\right)}\right)\\ & +|{\vec{b}}^{\left(1\right)}⊕{\vec{c}}^{\left(2\right)}\rangle \langle {\vec{b}}^{\left(1\right)}⊕{\vec{c}}^{\left(2\right)}|★\Gamma \left(\varrho_{\mathrm{BTN}}^{\left(B_{2}C_{1}\right)}\right)+\mathrm{diag}\left\{\Gamma \left(\varrho_{\mathrm{BTN}}^{\left(X_{1}\right)}\right)⊗\Gamma \left(\varrho_{\mathrm{BTN}}^{\left(X_{2}\right)}\right),X=A,B,C\right\}. \end{array}$$

Therefore, in order to test compatibility with the btn scenario for a given state, one can check if its cm can be written like the right-hand side of the above equation. While this might be cumbersome to test, we notice that the matrix $\Gamma_{\mathrm{BTN}}-R$ is also psd, which can also be used to check compatibility in the following way:

**Proposition** **3**(Positivity condition)**.**
*The matrix*
(19)$$\begin{array}{cc} \Xi (\varrho_{\mathrm{BTN}})= & \;\Gamma \left(\left\{G_{\alpha }^{A_{1}}⊗G_{\beta }^{A_{2}},G_{\gamma }^{B_{1}}⊗G_{\delta }^{B_{2}},G_{\epsilon }^{C_{1}}⊗G_{\zeta }^{C_{2}}\right\},\varrho_{\mathrm{BTN}}\right)\\ \; & -\mathrm{diag}\left\{\Gamma \left(\left\{G_{\alpha }^{X_{1}}\right\},\varrho_{\mathrm{BTN}}^{\left(X_{1}\right)}\right)⊗\Gamma \left(\left\{G_{\beta }^{X_{2}}\right\},\varrho_{\mathrm{BTN}}^{\left(X_{2}\right)}\right),X=A,B,C\right\} \end{array}$$
*is positive semi-definite.*

We note that neither term of the right-hand side of Equation (Equation 19) contains the reduced observables, which makes Ξ easy to compute.

An advised reader might point out that in order to verify if a given state is compatible with the btn scenario, it suffices to test whether $\varrho_{\mathrm{BTN}}=\varrho_{\mathrm{BTN}}^{\left(A_{2}B_{1}\right)}⊗\varrho_{\mathrm{BTN}}^{\left(B_{2}C_{1}\right)}⊗\varrho_{\mathrm{BTN}}^{\left(C_{2}A_{1}\right)}$, up to reordering of the subsystems. We stress that although this simple equation does answer the question, it requires the knowledge of the full density operator, whereas cm-based criteria only need expectation values of some chosen observables in order to be evaluated.

To close this section on btn, we present a few examples. First, we note that the lowest-dimensional achievable states are sixty-four-dimensional states (six qubits, or three ququarts (a ququart (sometimes ququad) is a four-dimensional quantum system)), and that the local dimensions cannot be prime numbers. Therefore, we start with the three-ququart ghz state,
(20)$$|GHZ_{4}\rangle =\frac{1}{\sqrt{2}}(|000\rangle +|333\rangle ),$$
which we mix with white noise
(21)$$\varrho_{GHZ_{4}}\left(v\right)=v|GHZ_{4}\rangle \langle GHZ_{4}|+\left(1-v\right)\frac{1_{64}}{64},$$
where *v* is the visibility. The corresponding Ξ matrix is psd only for p=0, meaning that the ghz state cannot be prepared in a btn network even with a very high amount of white noise. The same result is obtained when applied to the four-level ghz state, 12(|000〉+|111〉+|222〉+|333〉).

Proposition 3 may also be applied to three-ququart Dicke states, which are defined by
(22)$$|D_{3,4,k}\rangle =N\sum_{i_{1}+i_{2}+i_{3}=k}|i_{1}i_{2}i_{3}\rangle ,\;k=1,\ldots ,9,$$
with N being a normalisation factor. When mixed with white noise, they cannot be prepared in the btn scenario when p≠0 and p≠1 for k=1, and when p≠0 for k=2,…,7. More generally, by directly applying the result of Proposition 2, we can check whether the cm of a btn state can be written like the right-hand side of Equation (Equation 18). Performing this for $|D_{1}^{3}\rangle$, we conclude that this state cannot be generated in the btn scenario. On the other hand, the cm of the maximally mixed state 164 has such a decomposition.

The nature of interesting states that can be prepared in the btn scenario remains an open question. An obvious approach would be to distribute three Bell pairs across the network, resulting in a three-ququart genuine multipartite entangled triangle state. Getting ahead of the next sections, where it will be permitted to apply local transformations on systems *A*, *B*, and *C*, it is less straightforward to see what operations could be applied after distributing, for instance, Bell pairs.

## 5. Triangle Network with Local Operations

Let us now consider the situation where Alice, Bob, and Charlie can perform unitaries on their respective systems. As described in Section 2, the global state now reads
(23)$$\begin{array}{cc} \varrho_{\mathrm{UTN}}= & \left(U_{A}⊗U_{B}⊗U_{C}\right)\varrho_{\mathrm{BTN}}\left(U_{A}^{†}⊗U_{B}^{†}⊗U_{C}^{†}\right). \end{array}$$
First, we note that in general, for any set of observables $\left\{M_{i}\right\}$, any unitary *U*, and any state ϱ, there exists an orthogonal matrix *O* such that [23]
(24)$$\Gamma \left(\left\{M_{i}\right\},U\varrho U^{†}\right)=\Gamma \left(\left\{U^{†}M_{i}U\right\},\varrho \right)=O^{T}\Gamma \left(\left\{M_{i}\right\},\varrho \right)O.$$
Note that not all orthogonal transformations of cms correspond to a unitary transformation on the system. From that, we obtain the following proposition:

**Proposition** **4.***Consider the* cm *of a* utn *state with observables $A_{i}$, $B_{j}$, $C_{k}$ that only act on the systems A, B, C, respectively. There exist an orthogonal matrix $O=O_{A}⊕O_{B}⊕O_{C}$ and a* btn *state $\varrho_{\mathrm{BTN}}$ such that the* cm *of $\varrho_{\mathrm{UTN}}$ can be written as*(25)$$\Gamma_{\mathrm{UTN}}\equiv \Gamma \left(\left\{A_{i},B_{j},C_{k}\right\},\varrho_{\mathrm{UTN}}\right)=O^{T}\Gamma_{\mathrm{BTN}}O$$*with $\Gamma_{\mathrm{BTN}}$ as in Equation (Equation 9).*

Thus, the cm of $\varrho_{\mathrm{UTN}}$ can always be decomposed as a sum of positive semi-definite matrices with the following block decomposition
(26)$$\Gamma_{\mathrm{UTN}}=\underset{O^{T}T_{c}O}{\underset{︸}{\left(\begin{array}{ccc} \begin{array}{c} ■ \end{array} & \begin{array}{c} ■ \end{array} & 0\\ \begin{array}{c} ■ \end{array} & \begin{array}{c} ■ \end{array} & 0\\ 0 & 0 & 0 \end{array}\right)}}+\underset{O^{T}T_{b}O}{\underset{︸}{\left(\begin{array}{ccc} \begin{array}{c} ■ \end{array} & 0 & \begin{array}{c} ■ \end{array}\\ 0 & 0 & 0\\ \begin{array}{c} ■ \end{array} & 0 & \begin{array}{c} ■ \end{array} \end{array}\right)}}+\underset{O^{T}T_{a}O}{\underset{︸}{\left(\begin{array}{ccc} 0 & 0 & 0\\ 0 & \begin{array}{c} ■ \end{array} & \begin{array}{c} ■ \end{array}\\ 0 & \begin{array}{c} ■ \end{array} & \begin{array}{c} ■ \end{array} \end{array}\right)}}+\underset{O^{T}RO}{\underset{︸}{\left(\begin{array}{ccc} ■ & 0 & 0\\ 0 & ■ & 0\\ 0 & 0 & ■ \end{array}\right)}}.$$
We may also look at this situation by noticing that the cms of utn states can be written as
(27)$$\begin{array}{cc} \Gamma_{\mathrm{UTN}} & =\Gamma (\left\{U_{A}^{†}A_{i}U_{A},U_{B}^{†}B_{j}U_{B},U_{C}^{†}C_{k}U_{C}\right\},\varrho_{\mathrm{BTN}})\\ \; & =T_{c}^{U}+T_{b}^{U}+T_{a}^{U}+R^{U}, \end{array}$$
with $T_{c}^{U}$ being the cms of $\varrho_{\mathrm{BTN}}^{\left(A_{2}B_{1}\right)}$ with the following reduced observables
(28)$$\begin{array}{cc} A_{U,i}^{\left(2\right)}\equiv & {\left(U_{A}^{†}A_{i}U_{A}\right)}^{\left(2\right)}\\ = & \sum_{\alpha ,\beta }\mathrm{tr}\left(U_{A}^{†}A_{i}U_{A}\;G_{\alpha }⊗G_{\beta }\right)\mathrm{tr}\left(G_{\alpha }\varrho_{A_{1}}\right)G_{\beta } \end{array}$$
and $B_{U,j}^{\left(1\right)}$ built in the same way. The matrices $T_{b}^{U}$ and $T_{a}^{U}$ are defined similarly. The matrix $R_{A}^{U}$ is equal to $A^{U}-E_{A}^{U}-F_{A}^{U}$. One issue with this formulation is that, if one wants to test whether a given state is compatible with the utn scenario, the unitaries $U_{A}$, $U_{B}$, and $U_{C}$ and the state $\varrho_{\mathrm{BTN}}$ corresponding to the decomposition of $\varrho_{\mathrm{UTN}}$ are in general not known, thus there is no way to explicitly know the reduced observables.

We are now interested in triangle networks in which the local operations can be any quantum channel, i.e., no longer restricted to unitary operations. By making use of the Stinespring dilation theorem [28], we can show that ctn states also lead to cms that posses a block decomposition. As a matter of fact, any channel can be implemented by performing a unitary transformation on the system together with an ancilla, and then tracing out the ancilla. The covariance matrix of any state ϱ after a channel E (i.e., E(ϱ)) with observables $\left\{M_{i}\right\}$, therefore, has the same expression as taking the cm of the state together with an ancilla and applying the corresponding unitary *U*, that is, $U\left(\varrho ⊗\varrho_{\mathrm{ancilla}}\right)U^{†}$, with observables $\left\{M_{i}⊗1_{\mathrm{ancilla}}\right\}$. We can also see this by noticing that the cm of a reduced state is just a principal submatrix of the cm of the global state. Applying this to each node of the triangle network, we obtain the following proposition:

**Proposition** **5**(Block decomposition for cms of ctn states)**.**
*Let $\Gamma_{ctn}$ be the covariance matrix of $\varrho_{\mathrm{CTN}}=E_{A}⊗E_{B}⊗E_{C}\left(\varrho_{\mathrm{BTN}}\right)$ with local observables $A_{i}$, $B_{j}$, and $C_{k}$ as in Equation (Equation 5). There exist matrices $\Upsilon_{i}^{X}$ (X=A,B,C and i=1,2) such that*
(29)$$\Gamma_{\mathrm{CTN}}=\underset{\mathrm{PSD}}{\underset{︸}{\left(\begin{array}{ccc} \Upsilon_{2}^{A} & \gamma_{E} & 0\\ \gamma_{E}^{T} & \Upsilon_{1}^{B} & 0\\ 0 & 0 & 0 \end{array}\right)}}+\underset{\mathrm{PSD}}{\underset{︸}{\left(\begin{array}{ccc} \Upsilon_{1}^{A} & 0 & \gamma_{F}\\ 0 & 0 & 0\\ \gamma_{F}^{T} & 0 & \Upsilon_{2}^{C} \end{array}\right)}}+\underset{\mathrm{PSD}}{\underset{︸}{\left(\begin{array}{ccc} 0 & 0 & 0\\ 0 & \Upsilon_{2}^{B} & \gamma_{G}\\ 0 & \gamma_{G}^{T} & \Upsilon_{1}^{C} \end{array}\right)}}.$$

Comparing to Equation (Equation 26), we consider that we distributed the black blocks to the first three matrices. Although the proof techniques differ notably, the block decomposition has already been presented in the first work on cms of network states [11]. As demonstrated in that same work, Proposition 5 can be evaluated as an sdp. However, we are seeking practical analytical methods and criteria to determine if a state cannot be prepared in a network setting. In the next section, we present such a criterion that follows from Proposition 5.

Let us briefly discuss how this proposition applies to losr triangle network states, which can be expressed as $\varrho_{\Delta }=\sum_{\lambda }p_{\lambda }\varrho_{\mathrm{CTN}}^{\lambda }$. We recall that in this set up, the sources and the local operations may be coordinated by a classical random variable λ. From the concavity property in Ref. [23], we know that the difference between the covariance matrix $\Gamma (\varrho_{\Delta })$ and the weighted sum of cms $\sum_{\lambda }p_{\lambda }\Gamma \left(\varrho_{\mathrm{CTN}}^{\lambda }\right)$ is a psd matrix. Using Proposition 5, we directly obtain that there exist block matrices such that
(30)$$\Gamma \left(\varrho_{\Delta }\right)\geq \underset{\mathrm{PSD}}{\underset{︸}{\left(\begin{array}{ccc} \begin{array}{c} ■ \end{array} & \begin{array}{c} ■ \end{array} & 0\\ \begin{array}{c} ■ \end{array} & \begin{array}{c} ■ \end{array} & 0\\ 0 & 0 & 0 \end{array}\right)}}+\underset{\mathrm{PSD}}{\underset{︸}{\left(\begin{array}{ccc} \begin{array}{c} ■ \end{array} & 0 & \begin{array}{c} ■ \end{array}\\ 0 & 0 & 0\\ \begin{array}{c} ■ \end{array} & 0 & \begin{array}{c} ■ \end{array} \end{array}\right)}}+\underset{\mathrm{PSD}}{\underset{︸}{\left(\begin{array}{ccc} 0 & 0 & 0\\ 0 & \begin{array}{c} ■ \end{array} & \begin{array}{c} ■ \end{array}\\ 0 & \begin{array}{c} ■ \end{array} & \begin{array}{c} ■ \end{array} \end{array}\right)}}.$$
While a similar trick can lead to powerful necessary criteria for separability in the case of entanglement [23], it is not the case here. This is because the extreme points in the case of losr triangle network states are not well characterised, as already discussed in Supplementary Note 1 of Ref. [17]. Indeed, while it is known that the extreme points are of the form $E_{A}⊗E_{B}⊗E_{C}\left(|c\rangle c\langle |⊗|b\rangle b\langle |⊗|a\rangle a\langle |\right)$, it is not clear whether they are necessarily pure: on the one hand, the author of Ref. [13] showed that no three-qubit genuine multipartite entangled (gme) pure state can be generated in a triangle network, and on the other hand, the authors of Ref. [10] managed to find a state in the losr triangle network that has a fidelity to the ghz state of 0.5170. From Ref. [29], we know that states with such fidelity must be gme, thus, there exist extremal points of the set of three-qubit losr network states that are gme mixed states. Finally, we notice that pure gme states can exist in higher-dimensional triangle networks: For instance, the three-ququart state ${|\phi^{+}\rangle }_{A_{2}B_{1}}⊗{|\phi^{+}\rangle }_{B_{2}C_{1}}⊗{|\phi^{+}\rangle }_{C_{2}A_{1}}$ is gme for the partition $A_{1}A_{2}|B_{1}B_{2}|C_{1}C_{2}$ [16].

When additional properties of the states are known, such as the purity or the rank, sdp-based criteria for losr networks can be obtained, as shown in Ref. [25].

## 6. Covariance Matrix Criterion for Triangle Network States

As seen in the previous section, cms of ctn states with local observables $\left\{A_{i},B_{j},C_{k}\right\}$ possess a block decomposition. From Proposition 5, we obtain inequalities for any unitarily invariant norm ‖·‖,
(31)$$2\|\gamma_{E}\|\leq \|A_{2}\|+\|B_{1}\|,$$
for which we can take the trace norm and obtain
(32)$$2{\left\|\gamma_{E}\right\|}_{\mathrm{tr}}+2{\left\|\gamma_{F}\right\|}_{\mathrm{tr}}+2{\left\|\gamma_{G}\right\|}_{\mathrm{tr}}\leq \mathrm{tr}\left(A_{1}+A_{2}+B_{1}+B_{2}+C_{1}+C_{2}\right).$$
This gives us a direct necessary criterion for triangle network states.

**Proposition** **6**(Cm and trace norm criterion for triangle network states)**.**
*Let* Γ *be the* cm *of a triangle network state $\varrho_{c}tn=E_{A}⊗E_{B}⊗E_{C}\left(\varrho_{\mathrm{BTN}}\right)$ with local observables $\left\{A_{i},B_{j},C_{k}\right\}$. Then,*
(33)$$\mathrm{tr}(\Gamma )\geq 2{\left\|\gamma_{E}\right\|}_{\mathrm{tr}}+2{\left\|\gamma_{F}\right\|}_{\mathrm{tr}}+2{\left\|\gamma_{G}\right\|}_{\mathrm{tr}}$$
*has to hold, with $\gamma_{E}$, $\gamma_{F}$, and $\gamma_{G}$ as in Equation (Equation 5).*

Now, we apply the trace norm criterion to exclude states from the triangle network scenario.

First, we notice that, contrarily to btn states, three-qubit states can be generated in the ctn scenario. Therefore, we first consider the three-qubit ghz state that we mix with white noise, i.e.,
(34)$$\varrho_{GHZ}\left(v\right)=v|GHZ\rangle \langle GHZ|+\left(1-v\right)\frac{1_{8}}{8}.$$
By taking the three-qubit observable set ${𝓢}_{GHZ}=\left\{\sigma_{z}11,1\sigma_{z}1,11\sigma_{z}\right\}$ for the cm, the cm and trance norm criterion excludes $\varrho_{GHZ}\left(v\right)$ for v>12. The cm of the w state 13(|100〉+|010〉+|001〉) with observables ${𝓢}_{W}=\left\{\sigma_{x}11,\sigma_{y}11,1\sigma_{x}1,1\sigma_{y}1,11\sigma_{x},11\sigma_{y}\right\}$ excludes it for v>34 by the same method. Note that we omitted the tensor product signs for readability.

Further than that, we can put a bound on the fidelity of a triangle state to the ghz state. Consider an arbitrary state $\varrho =F|GHZ\rangle \langle GHZ|+\left(1-F\right)\tilde{\varrho }$, where $\langle GHZ|\tilde{\varrho }|GHZ\rangle =0$. From Proposition 6 with ${𝓢}_{GHZ}$, we show that *F* cannot be larger than $3-\sqrt{5}≃0.76$. We note that this result was already obtained in Ref. [17] using similar methods, and that by exploiting symmetries, this upper bound on the fidelity can be improved to 12≃0.71, which is, to our knowledge, the best analytical bound so far.

It is worth realising that upper bounds on the fidelity of triangle states to a given target state |Ψ〉 also hold in the case of losr networks. Indeed, losr network states $\varrho_{\Delta }$ are states that can be written as a convex combination of ctn states, and thus, $\max_{\varrho_{\Delta }}\langle \Psi |\varrho_{\Delta }|\Psi \rangle =\max_{\varrho_{ctn}}\langle \Psi |\varrho_{\mathrm{CTN}}|\Psi \rangle$. From this, we can conclude that from Proposition 6 it follows that any state with a fidelity to the ghz state higher than $3-\sqrt{5}≃0.71$ is excluded also from the losr triangle network scenario.

Before closing this section, a brief comment is in order. At first glance, it may seem that by only using the $\sigma_{z}$ correlations of a three-qubit state, we could exclude it from the set of losr network states and, thus, learn about its entanglement. However, this would be problematic because all separable three-qubit states are in the set of losr triangle network states, and all $\sigma_{z}$ correlations can be simulated by separable states. However, this is not the logic of the argument above: Proposition 6 puts a bound on the extremal points of losr triangle network states, which then by convexity holds for all losr states. In order to draw a conclusion for a given state, knowledge of the fidelity to some target state is required, which requires additional measurements than the $\sigma_{z}$ correlations alone.

## 7. Ncds Networks

In this section, we show that the block decomposition of covariance matrices of network states can also hold for larger networks. Indeed, if we consider networks where two nodes share parties from at most one common source (ncds networks), the triangle network results can be extended. Examples of such networks are networks with bipartite sources.

More explicitly, consider an *N*-node ncds network with a set of sources S. The number of sources is given by |S|, and each source s∈S is the set of nodes the source connects. Let $\Gamma_{\mathrm{NCDS}}$ be the cm of a global state of such a network with observables $\left\{A_{x|i}:x=1,\ldots ,N\right\}$, where $A_{x|i}$ is the *i*th observable that only acts on the node *x*. Then, $\Gamma_{n}cds$ has a block form analogous to Equation (Equation 5), where the diagonal blocks are labelled $\Gamma_{x}$ and the off-diagonal block are $\gamma_{xy}=\gamma_{yx}^{T}$ (x≠y, x,y∈{1,…,N}). Formally, we state that

**Proposition** **7**(Block decomposition for cms of ncds network states)**.** *There exist matrices $\Upsilon_{x}^{s}$ (s∈S, x∈s) such that $\Gamma_{\mathrm{NCDS}}$ can be decomposed as a sum of |S| positive semi-definite block matrices $T_{s}$ (s∈S) where the off-diagonal blocks of each $T_{s}$ are $\gamma_{xy}$ for {x,y}⊂s and 0 for {x,y}⊄s, and where the diagonal blocks are $\Upsilon_{x}^{s}$.*

For a technical proof, see Appendix E. In there, we prove that in the case of basic (i.e., without local operations) networks with no common double source (bncds networks), the proposition holds. Following a similar line of reasoning to the proofs for triangle networks, the proposition naturally extends to ncds networks with local operations.

Let us consider an easy example for the sake of clarity. Figure 2 shows a five-partite network consisting of two tripartite sources $\varrho_{a}$ and $\varrho_{b}$, and one bipartite $\varrho_{c}$. The set of sources is given by S={a,b,c}={{1,2,3},{3,4,5},{1,5}}.

Following the notation of Proposition 7, there must exist eight matrices $\Upsilon_{1}^{a}$, $\Upsilon_{2}^{a}$, $\Upsilon_{3}^{a}$, $\Upsilon_{3}^{b}$, $\Upsilon_{4}^{b}$, $\Upsilon_{5}^{b}$, $\Upsilon_{1}^{c}$, and $\Upsilon_{5}^{c}$ such that the cm of the global network state
(35)$$\varrho_{\mathrm{NCDS}}=E_{1}⊗E_{2}⊗E_{3}⊗E_{4}⊗E_{5}\left(\varrho_{a}⊗\varrho_{b}⊗\varrho_{c}\right)$$
may be decomposed as
(36)$$\Gamma_{\mathrm{NCDS}}=\underset{T_{a}}{\underset{︸}{\left(\begin{array}{ccccc} \begin{array}{c} \Upsilon_{1}^{a} \end{array} & \begin{array}{c} ■ \end{array} & \begin{array}{c} ■ \end{array} & 0 & 0\\ \begin{array}{c} ■ \end{array} & \begin{array}{c} \Upsilon_{2}^{a} \end{array} & \begin{array}{c} ■ \end{array} & 0 & 0\\ \begin{array}{c} ■ \end{array} & \begin{array}{c} ■ \end{array} & \begin{array}{c} \Upsilon_{3}^{a} \end{array} & 0 & 0\\ 0 & 0 & 0 & 0 & 0\\ 0 & 0 & 0 & 0 & 0 \end{array}\right)}}+\underset{T_{b}}{\underset{︸}{\left(\begin{array}{ccccc} 0 & 0 & 0 & 0 & 0\\ 0 & 0 & 0 & 0 & 0\\ 0 & 0 & \begin{array}{c} \Upsilon_{3}^{b} \end{array} & \begin{array}{c} ■ \end{array} & \begin{array}{c} ■ \end{array}\\ 0 & 0 & \begin{array}{c} ■ \end{array} & \begin{array}{c} \Upsilon_{4}^{b} \end{array} & \begin{array}{c} ■ \end{array}\\ 0 & 0 & \begin{array}{c} ■ \end{array} & \begin{array}{c} ■ \end{array} & \begin{array}{c} \Upsilon_{5}^{b} \end{array} \end{array}\right)}}+\underset{T_{c}}{\underset{︸}{\left(\begin{array}{ccccc} \begin{array}{c} \Upsilon_{1}^{c} \end{array} & 0 & 0 & 0 & \begin{array}{c} ■ \end{array}\\ 0 & 0 & 0 & 0 & 0\\ 0 & 0 & 0 & 0 & 0\\ 0 & 0 & 0 & 0 & 0\\ \begin{array}{c} ■ \end{array} & 0 & 0 & 0 & \begin{array}{c} \Upsilon_{5}^{c} \end{array} \end{array}\right)}},$$
where the off-diagonal blocks are simply the ones from $\Gamma_{\mathrm{NCDS}}$.

We see directly that we can extend Proposition 6 as

**Proposition** **8**(Cm and trace norm criterion for ncds network states)**.**
*Let $\Gamma_{\mathrm{NCDS}}$ be as above. Then,*
(37)$$\mathrm{tr}(\Gamma_{\mathrm{NCDS}})\geq 2\sum_{x>y=1}^{N}{\left\|\gamma_{xy}\right\|}_{\mathrm{tr}}$$
*has to hold.*

We note that this criterion does not take network topology into account: it treats a network with a single N−1-partite source the same way it treats a line network with N−1 bipartite sources. While this is interesting as it can exclude states from all networks, we also expect it to be weaker than criteria designed for specific network topologies. On top of that, Proposition 8 only takes into account that the principal submatrices of each $T_{s}$ are positive semi-definite, not that the matrices themselves are psd.

As an example, let us consider an *N*-qubit ghz state, |GHZN〉=12(|0〉⊗N+|1〉⊗N), of visibility *v* mixed with white noise. As observables, we take $\sigma_{z}^{\left(x\right)}$ for each qubit *x*. The resulting cm will have diagonal elements equal to one, whereas the off-diagonal elements will be *v*. Applying the previous proposition, we exclude *N*-partite ghz states mixed with white noise from any ncds network scenario for
(38)$$v>\frac{1}{N-1}.$$
With N=3, we recover the result of the example for the triangle network.

Nevertheless, we are forced to observe that the criterion only considers two-body correlation, therefore, cannot fully capture the entanglement in the target states. To see this, let us look at the four-qubit cluster state |Cl〉=|+0+0〉+|+0−1〉+|−1−0〉+|−1+1〉 (up to normalisation). Its generators are $\sigma_{x}\sigma_{z}11$, $\sigma_{z}\sigma_{x}\sigma_{z}1$, $1\sigma_{z}\sigma_{x}\sigma_{z}$, and $11\sigma_{z}\sigma_{x}$, where the only two-body correlations are given by $\sigma_{x}\sigma_{z}11$ and $11\sigma_{z}\sigma_{x}$. A possible set of observables is $𝓢=\{\sigma_{x}^{\left(1\right)},\sigma_{z}^{\left(2\right)},\sigma_{z}^{\left(3\right)},\sigma_{x}^{\left(4\right)}\}$ and we obtain
(39)$$\Gamma \left(𝓢,|Cl\rangle \right)=\left(\begin{array}{cccc} 1 & 1 & 0 & 0\\ 1 & 1 & 0 & 0\\ 0 & 0 & 1 & 1\\ 0 & 0 & 1 & 1 \end{array}\right).$$
The trace criterion is satisfied and, thus, we cannot exclude |Cl〉 from ncds network scenarios by means of Proposition 8. Moreover, we directly see that the matrix has a block decomposition, namely, $\left(\begin{array}{cc} 1 & 1\\ 1 & 1 \end{array}\right)⊕\left(\begin{array}{cc} 1 & 1\\ 1 & 1 \end{array}\right)$, which a priori could arise from a network with two bipartite sources. However, we know from Ref. [17] that the four-qubit cluster state cannot be generated in bipartite networks.

## 8. Conclusions

In this work, we presented alternative proofs to the block decomposition of covariance matrices for network states. From these, we derived analytical criteria to certify that some states cannot be generated through quantum networks as we define them in Equation (Equation 3). This means that those excluded states either require global sources that connect all nodes, classical communication, non-local operations, or shared randomness to be generated. Concerning the latter resource, we also showed that Propositions 6 can be used to upper-bound the fidelity to some target states that losr network states can have. Furthermore, we stress that our criteria are analytical and computable.

Regarding extensions of our work, it would be worthwhile to investigate whether the proof of Proposition 7 can be extended to networks beyond ncds networks. As shown in Ref. [11], the latter is indeed possible, which implies that Propositions 7 and 8 hold for general networks as well.

Finally, as the field of network entanglement and its potential applications in quantum information theory continues to grow, it may be valuable to investigate additional avenues for identifying compatible network states. Specifically, an area of interest is finding sufficient criteria for network states, as current results only provide necessary criteria. By developing such criteria, we may be able to learn more about states that *can* be generated in networks without communication and about their potential usefulness, for instance, for quantum conference key agreement. In this context, it is interesting to also consider noisy networks: this would translate to imposing additional conditions on the sources states, e.g., by making them travel through depolarisation channels or by constraining their purity.

## Figures and Tables

**Figure 1 entropy-25-01260-f001:**
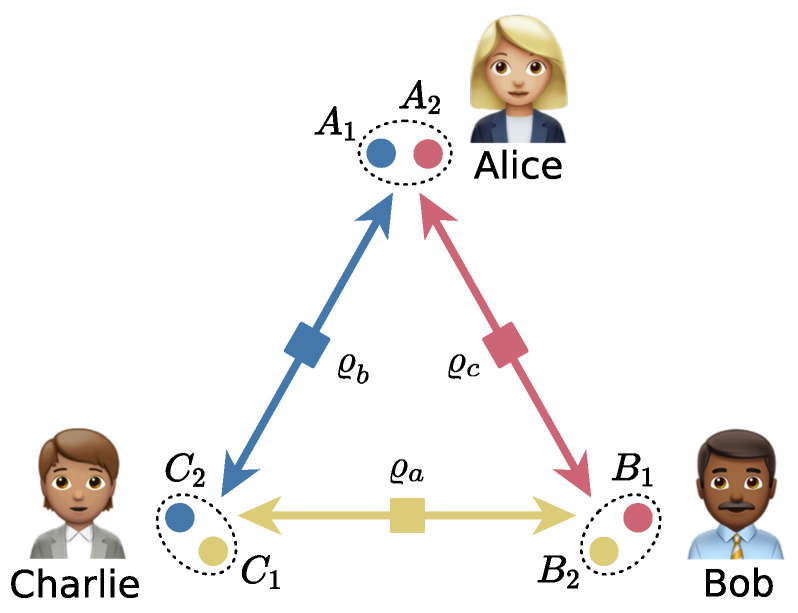
Basic triangle network. Each source distributes subsystems to the three nodes, Alice, Bob, and Charlie. They each end up with a bipartite system $X=X_{1}X_{2}$ (X=A,B,C), on which they could apply a local operation.

**Figure 2 entropy-25-01260-f002:**
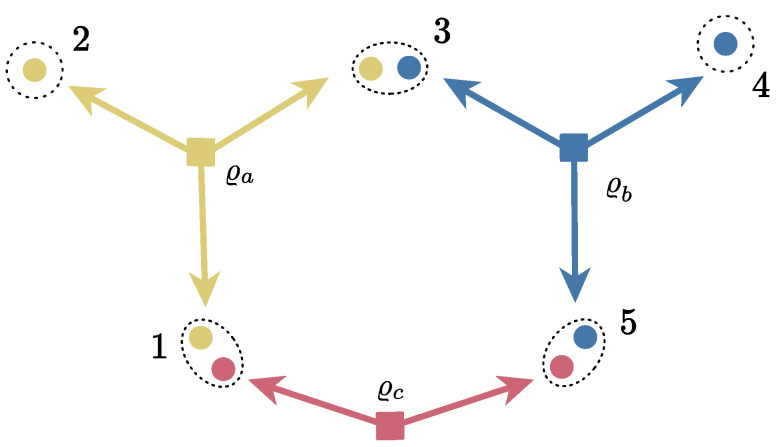
Five-partite network with two tripartite sources $\varrho_{a}$ and $\varrho_{b}$, and one bipartite source $\varrho_{c}$. The parties 1, 2, 3, 4, and 5 may perform a local channel $E_{i}$ on their system *i* (i=1,…,5).

## Data Availability

Not applicable.

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
