# Peer review of "Covariance-Matrix-Based Criteria for Network Entanglement"

_entropy, 2023, doi:10.3390/e25091260_

Round 1

Reviewer 1 Report

This manuscript is in my view very well written and pedagogical. It provides nice results, and I recommend publication in Entropy.

I have only a couple of very minor remarks that I suggest that the authors can do with as they please.

I believe that it could be useful if they authors could add a bit more of indications in the bulk-text concerning what results are old and only included for completeness and pedagogical purposes, what results are novel proofs to old results, and what results are new. I would not suggest any big additions, but I believe that some small indications in the running text could be very helpful to give the reader a clear picture of the nature of the contributions in this paper. In general, it could maybe also be useful, if possible, with a bit more explicit comparisons with the existing literature concerning the specific results, like what do the new results achieve that the previous results could not.

It seems to me that some of the proof-techniques employed presumes that the underlying Hilbert-spaces are finite-dimensional. This is a perfectly fine restriction to make, and is also explicitly stated in section 2. However, since this underlying assumption is not explicitly `declared' in the Propositions, it might easily be missed by a casual reader. It may be an overkill to say this in every relevant proposition, but maybe add one or two reminders somewhere in the text?

* In equation (A33), I guess that there are parentheses missing.

Author Response

Dear Reviewer,

We thank you for the careful reading of our manuscript.

Concerning your suggestions, we added the following paragraph after our Proposition 5:

Although the proof techniques differ notably, the block decomposition has already been presented in the first work on CMs of network states [11]. As demonstrated in that same work, Proposition 5 can be evaluated as an SDP. However, we are seeking practical analytical methods and criteria to determine if a state cannot be prepared in a network setting: In the next section, we present such a criterion that follows from Proposition 5.

and added a clarification about finite-dimensionality after our Proposition 1:

We want to emphasize that the results presented in this manuscript are valid only in the context of finite-dimensional Hilbert spaces. A potential future research direction is to investigate how these results can be extended to the infinite-dimensional case. As mentioned in the introduction, CMs are also well-suited for continuous variable systems.

The parentheses have been fixed.

With our best regards,

Kiara Hansenne and Otfried Gühne.

Reviewer 2 Report

In the present manuscript, the authors derive covariance matrix-based criteria for network entanglement. The work mainly focuses on a tripartite triangular network, but some results are extended also to more general networks. The authors consider network states that are obtained by sharing bipartite entangled states between pairs of nodes, possibly combined with local unitary operations or local quantum channels. The authors utilize covariance matrices, which are shown to possess specific structure for the considered class of states, in particular they can be expressed as a sum of block diagonal positive-semidefinite matrices. Based on this the authors derive analytical criteria, that any state belonging to the considered class of states must satisfy. The results represent an interesting contribution to the theory of multipartite quantum correlations.

I have two questions:

1)      Would it be possible to generalize the obtained results to infinite-dimensional Hilbert spaces, in particular to scenario where qudits are replaced with modes of electromagnetic field? In such continuous variable scenario, covariance matrices of quadrature operators are of particular interest. Could we obtain some interesting information about the structure of a multimode multipartite CV quantum state using its covariance matrix of quadrature operators and the techniques of the present manuscript?

2)      The authors mention that classical communication among the network nodes is not allowed. This is very important, because if classical communication is allowed then any multipartite state can be distributed over the network using shared bipartite entanglements and quantum teleportation. Therefore, the lack of classical communication is crucial to make the problem studied in the present manuscript nontrivial. In quantum communication, the classical communication is normally considered to be cheap and available. The authors should provide a more detailed explanation an motivation why it is of interest to consider a scenario where the classical communication (combined with local quantum operations determined by the classical messages) is forbidden or inaccessible.

Author Response

Dear Reviewer,

We thank you for the careful reading of our manuscript.

Here are the replies to your questions:

  1. Concerning continuous variable systems, we do agree that it is an interesting question to see how or whether the results extend to the infinite-dimensional case. However, we believe that it is outside of the scope of our paper. We nevertheless added the following paragraph right after our first Proposition:

We want to emphasize that the results presented in this manuscript are valid only in the context of finite-dimensional Hilbert spaces. A potential future research direction is to investigate how these results can be extended to the infinite-dimensional case. As mentioned in the introduction, CMs are also well-suited for continuous variable systems.

  1. To address the second remark, we have added this paragraph at the beginning of Section 2:

In addition, the parties in the triangle network considered here do not have access to classical communication, which prevents them from executing teleportation or entanglement swapping protocols. Although classical communication is usually considered a cheap resource in quantum communication protocols, it does require time. The classical information must be communicated across the network, which can introduce undesirable latency, particularly in a context where quantum memories are still sub-optimal and expensive.

We hope this answers your questions.

With our best regards,

Kiara Hansenne and Otfried Gühne.

Reviewer 3 Report

entropy-2541132-peer-review-v1

Covariance matrix-based criteria for network entanglement

The authors study the decomposition of covriance matrices under the network scenario, in particular to a tripartite system, as a sum of positive semidefinite block matrices. They have also derived some results to certify if states can be generated through quantum network. The paper is technically correct and deserves publication with minor amendments.  However, before acceptance, the paper should preferably be revised as follows:

(i) Notations are a little confusing at times. 

(ii) At the end of section 4, it is shown that both GHZ state and the Dicke state cannot be prepared as BTN states and that the only BTN decomposable state is the trivial maximally mixed state. Perhaps, the authors might wish to describe another not-so-trivial basic triangle network (BTN) decomposable state.

(iii) Can the authors say what happens in the case of noisy networks?

Minor comments

The matrices $T_x$ in Eq(9) do not appear in the equation.   Please amend. There is also confusion in the notation of $\{ A_i \}$. 

Only minor changes needed.

Author Response

Dear Reviewer,

We thank you for the careful reading of our manuscript.

We added the following paragraph at the end of Section 4:

The nature of interesting states that can be prepared in the BTN scenario remains an open question. An obvious approach would be to distribute three Bell pairs across the network, resulting in a three-ququart genuine multipartite entangled triangle state. Getting ahead of the next sections where it will be permitted to apply local transformations on systems A,B and C, it is less straightforward to see what operations could be applied after distributing for instance Bell pairs.

Concerning noisy networks, we understand that the question is the following: What happens if the sources states travel through a noisy channel? The final state of a triangle network would read EA ⊗ EB ⊗ EC [ E(rhob) ⊗ E(rhoc) ⊗ E(rhoa) ], where E would be the noisy channel. This would correspond to putting restrictions on the source states, e.g. saying that the sources have to be of the form of (1-p) |Psi><Psi| + p 1/d, which would further reduce the set states achievable in networks. Although this is an interesting consideration, we believe that it is outside of the scope of our paper, as we do not think that the decomposition of CMs into block positive semidefinite matrices would be a suitable tool to discuss this. We added a few sentences about this in the conclusion, as a future research direction.

The use of T_x has been corrected and the use of \{A_i\} has been clarified.

We hope this answers your questions and thank you again for your remarks.

With our best regards,

Kiara Hansenne and Otfried Gühne.

Round 2

Reviewer 2 Report

The authors have satisfactorily responded to the questions raised in my previous report and the revised manuscript is suitable for publication. I recommend publication in its present form.

Reviewer 3 Report

No further comments.  The revised manuscript can be published as it stands.